# Proposal of Sustainability Indicators for the Design of Small-Span Bridges

**DOI:** 10.3390/ijerph17124488

**Published:** 2020-06-22

**Authors:** Cleovir José Milani, Víctor Yepes, Moacir Kripka

**Affiliations:** 1Department of Civil Engineering, Federal Technological University of Paraná, Via do Conhecimento, Km 1, Pato Branco, Paraná 80000, Brazil; milani@utfpr.edu.br; 2Institute of Concrete Science and Technology (ICITECH), Universitat Politècnica de València, 46022 Valencia, Spain; 3Civil and Environmental Engineering Graduate Program, University of Passo Fundo, BR 285, São José, Passo Fundo, Rio Grande do Sul 90000, Brazil; mkripka@upf.br

**Keywords:** bridges, sustainability, design, life cycle assessment

## Abstract

The application of techniques to analyze sustainability in the life cycle of small-span bridge superstructures is presented in this work. The objective was to obtain environmental and economic indicators for integration into the decision-making process to minimize the environmental impact, reduce resource consumption and minimize life cycle costs. Twenty-seven configurations of small-span bridges (6 to 20 m) of the following types were analyzed: steel–concrete composite bridges, cast in situ reinforced concrete bridges, precast bridges and prestressed concrete bridges, comprising a total of 405 structures. Environmental impacts and costs were quantified via life cycle environmental assessment and life cycle cost analysis following the boundaries of systems from the extraction of materials to the end of bridge life (“from cradle to grave”). In general, the results indicated that the environmental performance of the bridges was significantly linked to the material selection and bridge configuration. In addition, the study enabled the identification of the products and processes with the greatest impact in order to subsidize the design of more sustainable structures and government policies.

## 1. Introduction

Bridges have a major role in transportation infrastructure, supporting highway traffic loads, crossing various obstacles and enabling effective communication between two destinations [1]. Bridges generally represent a significant public resource; in Europe, for example, bridges account for approximately 2% of the road network length and 30% of its cost [2].

Bridges should be seen as a key part of the economic activity and well-being of a given community [3]. While a bridge is designed from economic, technical and safety perspectives, the environmental performance is often not considered in the decision-making process [4]. Several elements should be considered in the bridge life analysis process: factors concerning age, which are directly related to maintenance and various rehabilitation, repair or reinforcement procedures, and others related to the increasing weight of trucks and transportation facilities that pass over the bridge [5].

Given the importance of bridges and their relationship with the environment, the pillars of sustainability must be observed, meaning that the economic pillar of sustainable development must ensure the efficiency of natural resource consumption [3]. The potential environmental impacts must be measured by indicators because the sustainability of bridges includes the extension of maintenance and the end of the bridge’s life [6]. The social pillar must be observed considering its positive and negative impacts [7].

The environmental performance of bridges is directly related to the choice of construction materials as well as to the type of bridge [8,9,10]. Currently, the use of sustainable materials in the construction of bridges is increasingly attempted considering economic, social and environmental factors [11]. Life cycle sustainability assessments applied to small-span bridge superstructures for side roads have revealed the good performance of composite steel and concrete structures (concrete deck and steel profile beams) [12].

The sustainable use of raw materials should be an objective of the construction industry [13], with emphasis on steel/concrete composite bridges (concrete deck and steel profile beams) [14]. Regardless of the criteria used to represent the sustainability of the structures, decision-making processes should include a complete analysis of the life cycle, from the cradle to the grave [15], considering the possibility of abiotic resource depletion, the acidification of the environment, the depletion of the stratospheric ozone layer, the eutrophication of water bodies and the photochemical creation of ozone [16].

With regard to life cycle cost analysis (LCCA), environmental, economic and social indicators [17] can be applied as a tool for calculating and comparing the life cycle cost of a product [18]. This allows LCCA to be interpreted as one of the three pillars of sustainability [19], and it is fundamental to the structure of sustainability due to its usefulness for determining the cost-effectiveness/cost-competitiveness ratio of various technological options that affect the environment [20].

The LCCA may include the total life cycle time, which takes into account the costs associated with the different phases through which the construction work proceeds over time [21], and the user’s costs. It is characterized as total costs and evolves rapidly, meaning that the assessment of the structural performance and total accumulated cost throughout the life cycle determines the competent management of civil infrastructure [22].

The largest share of bridges is represented by small-span structures. For example, in a study conducted in Brazil, it was found that 74% of the bridges had spans smaller than 10 m, and 90% had lengths of up to 20 m [12]. Apparently, small-span bridges are generally underestimated considering the lack of executive design and technical monitoring in their construction, non-existent maintenance, evident signs of abandonment including poor conditions for use and the emergence of different pathologies or a lack of attention from the public authorities, thus generating sustainability issues [23]. In this regard, the rationalization of the process of building structures—including reducing costs [24,25,26], social impacts [27,28] and environmental impacts [29,30]—should be addressed in the design phase [31]. The environmental impact of bridges has been highlighted due to the consumption of resources [32]; it has been widely demonstrated that more sustainable construction benefits humanity [33].

The aim of this study was to determine the economic and environmental indicators related to bridge sustainability that should be incorporated into the decision-making process of design to alleviate environmental impacts, reduce resource consumption and minimize costs throughout the life cycle. Several small-span bridge configurations were analyzed, with their potential environmental impacts and costs quantified by life cycle environmental assessment and life cycle cost analysis, following the boundaries of systems from the extraction of materials to the end of bridge life (i.e., “from cradle to grave”).

The study was performed using life cycle assessment (LCA) and life cycle cost analysis (LCCA), using the guidelines of ISO 14040 [34] and ISO 14044 [35]; the International Reference Life Cycle Data System (ILCD) [36]; the Ecoinvent 3 database. 5, 2018 [37]; the ReCiPe 2016 V1.1 global reach method [38]; the SimaPro software, version 9.0.0.32 [39,40]; the European methods IMPACT 2002 + V2.14 [41] and ILCD 2011 Midpoint + V1.10 [42]; and the American method Building for Environmental and Economic Sustainability (BEES + V4. 07 USA) [43].

Comparative analysis was carried out using the following the methods: the European method IMPACT 2002 + V2.14, which was formulated by the Swiss Federal Institute of Technology in Lausanne (EPFL) and proposes the combination of midpoint and endpoint approaches, characterizing the inventory results into 15 impact categories correlated to at least one of the four categories of damage [41]; the ILCD 2011 Midpoint + V1.10 method, which was launched by the European Commission, Joint Research Centre, in 2012 [42]; and the US method BEES + V4.07 USA, which measures the environmental performance of building products by using the life-cycle assessment approach specified in the ISO 14040 series of standards [43].

In this process of building sustainable bridges, the terms “sustainable development,” “sustainability” and “sustainable” were cited in the Brundtland Report [44] in the context of economic, social and environmental issues [45], which attributed the genesis of terminology to the document registered by Hans Carl von Carlowitz in 1713 [46]. The life cycle assessment (LCA) method by Boulenger [47], in which some aspects of the assessment of sustainability [48] in bridge design were shown that aim at achieving a low impact on the life cycle [49], was also presented, observing the ecological, economic and social pillars of sustainability [50] with a management strategy that seeks to equalize the cost of annual maintenance for a stable budget [51].

The European Union formulated the three pillars of sustainability at the Copenhagen Summit and the Treaty of Amsterdam, 1997, forming a basis on which to build sustainable development [50], the performance of which is measured through environmental life cycle analysis, life cycle cost analysis and social life cycle analysis [52]. Regarding social analysis, there are a variety of criteria that indicate a lack of consensus and for which there is less certainty and discussions regarding the selection of more representative criteria are triggered [53]. Some of the important aspects of sustainability are associated with the extraction of construction raw materials, having direct positive and negative impacts on the environment, the economy and the social context as a whole [7].

The focus on sustainability in bridge design lies in the investigation of economic and environmental impacts with respect to the analysis of materials and the concept of sustainable development in the bridge industry [54]. The sustainability LCA of bridge structures is recommended to include the environmental, economic, social and functional qualities of bridges [55], although Hammervold, Reenaas and Brattebo [56] have proposed the inclusion of more categories, which impact the environment for comparative analyses.

Studies on the analyses of the life cycle of bridges have sought the explanation of the uncertainties associated with bridge maintenance [57,58,59], with reference to the application of holistic approaches in studies related to sustainable development in the bridge industry [60,61]. This view is different from that presented by Du et al. [62], who identified the variable influences of impact categories resulting from materials, structural elements and general design.

Tapia [63] evaluated the sustainability life cycle of deteriorated bridges to assess risk-based sustainability indicators of bridge performance. Pang et al. [64] had previously verified the issues related to old bridges in China, most of which were in need of maintenance due to the aging of road bridge materials over time, which has significant effects on structural performance [65].

Regarding the state of the art of sustainability and its pillars, the scarcity of studies on sustainable processes in bridge designs was considered as an opportunity to seek answers and to formulate alternatives for bridges that meet the needs of the population in their free transit and that can be built based on the parameters of sustainability.

Almeida, Teixeira and Delgado [66] developed a method including tests with combined approaches as a model for minimizing bridge life cycle costs, aiming at the optimization of maintenance intervention plans in a set of concrete bridges to calculate deterioration over time and to analyze costs. Zhang, Wu and Wang [9] assigned probability distributions for the parameters considered in the environmental impact criteria, evaluating the variability in inventory acquisition and detecting the key parameters with relevant environmental impacts. Penadés-Plá et al. [11] used several methods and sustainable criteria for decision-making in each phase of a bridge’s life cycle, ranging from design to the recycling or demolition of the structure.

For existing bridges—specifically, steel/concrete composite bridges—the study by Bizjak et al. [67] indicated that the use of concrete reinforced with ultra-high-performance fiber in the construction and composition of the slab minimizes problems related to the adjustment of the sub-structure geometry, reducing the use of resources.

Huijbregts et al. [68] recommended the ReCiPe method as it provides a harmonized implementation of cause and effect paths for the calculation of the characterization factors of potential environmental impacts and damages (end point).

Although this study has a clear objective of proposing sustainability indicators for the economic and environmental pillars in the life cycle of small-span bridge superstructures, a brief statement is needed on the decisive issue of the decision-making process for the choice of small-span bridges with better sustainability performance. The use of a decision-making process enables solutions that meet normally conflicting objectives to be achieved, and this process can be conducted in various ways.

In the study undertaken by Kripka, Yepes and Milani [69] and the data collected at the beginning of this study, sustainable design alternatives for small-span bridges in Brazil were investigated. In this approach, the structures were evaluated while taking into account quantitative aspects (construction cost, assembly, materials transport, lifespan and environmental impact) and qualitative aspects (architecture and sense of safety for the user). Decision-making methods with several criteria were applied, minimizing the subjectivity implicit in the decision-making process. In this study, two decision-making methods with several criteria were adopted: the analytical hierarchy process (AHP) and the VIKOR method.

Many sustainability indicators have been applied in studies on bridges, in which the authors have chosen both the method and the indicators. Padgett and Tapia [70] applied a risk-based method for life-cycle environmental sustainability analysis. Du et al. [4] analyzed 20 environmental indicators presented in five bridge designs. Tapia [63] addressed the issue of deteriorated bridges with the objective of quantifying risk-based sustainability indicators of performance. Du highlighted global warming and energy consumption as two popular indicators in LCA [8]. Arya, Amiri and Vassie [71] used indicators such as climate change, resource use, waste, biodiversity and heritage. Hatami and Morcous [72] presented a proposal to undertake a life cycle cost analysis. Furuta used the Genetic Algorithm to minimize the total life cycle cost of a large number of bridges [51]. There is a perception that there is a lack of a standardized bridge life cycle assessment manual that can guide designers in their decision-making process for choosing the best design, with the goal of minimizing environmental and economic damage and a focus on sustainability.

In the construction of a bridge, a comparative study between various solutions must be carried out, with the final choice of a solution being made when all items of functionality, safety, esthetics, economy and durability are met [73]. For this research work, it was considered relevant to insert one more requirement; i.e., bridge sustainability indicators. The selection of the indicators must be in accordance with the objectives of the study, which require previous knowledge of the life cycle of the product or processes analyzed. Life cycle impact assessment (LCIA) provides the results of impact indicators related to human health, the natural environment and resource depletion [74].

There is an increase of global interest in environmental issues in general, and a growing number of studies aim to identify alternatives to improve production processes and the use of common resources. Although only few studies have been related to small-span bridges, significant findings were determined relative to bridges, as this topic concerns public equipment, which is relevant to society in daily use and is equally important for the purposes of sustainability when investigating the possibilities of the sustainable development of bridges in the design phase.

These possibilities can guide the development of bridge design based on the pillars of sustainability, enabling a study that analyzes bridge conditions by applying the available tools and resources to provide the opportunity to create alternatives to reduce environmental impacts.

The remainder of this paper is structured as follows. Section 2 introduces the proposed methodology and Section 3 presents results and discussions. Finally, in Section 4, the conclusions and final considerations are presented.

## 2. Materials and Methods

This section presents the methodology for the application of the LCA and LCCA techniques for small-span bridge superstructures, which are designed with different typologies and materials to identify their potential environmental impacts and characterize their effects on human health, ecosystems and resources and their costs in the life cycle of the bridges.

In recent years, many methods of impact characterization have been developed; examples are methods such as ILCD 2011 MIDPOINT + and IMPACT 2002 + in Europe and BEES and TRACI 2.1 in North America. However, the method that has become prominent for researchers is the global ReCiPe method (adopted in this work).

In this study, the application of the LCA on bridges was based on ISOs 14040 and 14044 and the ILCD guidelines, using the Ecoinvent database 3.5 from August 2018, the ReCiPe (H) 2016 V1.1 global reach method, which is a harmonized method for assessing the impact of the life cycle at the endpoint level, and the SimaPro software version 9.0.0.32. The results presented in this study reference damage to human health, ecosystems and resources.

ISO 14040 sets out the basic safety inspection guidelines for environmental management, life cycle assessment and their principles and framework. ISO 14044 deals with environmental management, life cycle assessment and requirements and guidelines, and ILCD provides a common basis for consistent, robust and quality-assured life cycle data and studies. Such data and studies support consistent sustainable consumption and production tools such as environmental labeling, eco-design, carbon footprint reduction and green public procurement.

Ecoinvent is a large database of LCIA in which data are transparently documented as inputs/outputs in the unitary process. ReCiPe is a method for life cycle impact assessment, and the primary objective of the ReCiPe method is to transform the long list of life cycle inventory results into a limited number of indicator scores. These indicator scores express the relative severity for an environmental impact category. In ReCiPe, indicators can be obtained at two levels: 18 midpoint indicators and three endpoint indicators. SimaPro, which is a type of software used for life cycle assessment (LCA), has the function of collecting data and analyzing the environmental performance of products and services and can model and analyze complex life cycles in a systematic and transparent way, following the recommendations of the ISO 14040 series.

At the damage assessment stage, impact category indicators with a common unit can be added. In the ReCiPe method, the 18 impact categories are listed under three endpoint damage categories:Damage to human health: This is expressed as the number of years of life lost and the number of years of living with disability. These are combined as Disability Adjusted Life Years (DALYs)—an index that is also used by the World Bank and World Health Organization. Regarding the end point category of human health damage, the following items can be found at the mid-point that impact human health: global warming, stratospheric ozone depletion, ionizing radiation and ozone formation, fine particulate formation, human carcinogenic toxicity, non-human carcinogenic toxicity and water consumption.Damage to ecosystems: This is expressed as the loss of species over a certain area for a certain time; the unit is species per year. Regarding the category of final ecosystem damage, the following can be found at an average level that impacts various ecosystems: global warming and terrestrial and freshwater ecosystems, and ozone formation and terrestrial ecosystems. In addition, terrestrial and aquatic ecosystems are affected by terrestrial acidification, eutrophication of freshwater and marine water, ecotoxicity of terrestrial, freshwater and marine ecosystems, land use and water consumption.Resource scarcity: This is expressed as the excess costs of future resource production over an infinite period (assuming constant annual production), considering a discount ratio of 3%; the unit is USD2013. The endpoint damage in the resource category refers to the scarcity of mineral resources, as an average impact.

Furthermore, to validate the results obtained in the ReCiPe methodology, the sensitivity analysis was performed with the application of the following methods for characterization of environmental impacts: the European methods IMPACT 2002 + V2.14 and ILCD 2011 Midpoint + V1.10, and the American method BEES + V4.07 USA.

Initially, the objective of the LCA study was defined as follows: the intended application of the LCA results should be declared in a precise and unequivocal manner, along with the assumed data and methodological limitations of the study, reasons for conducting the LCA study, identification of the target audience of the study, whether the LCA study includes a comparative statement to be disseminated to the public, to whom the results of the study are intended to be communicated, the actors involved and the identification of those who commissioned the LCA/CAV study [74]. Next, the scope, inventory preparation, analysis and impact assessment were defined.

All stages were analyzed and interpreted considering steel/concrete composite bridges, reinforced concrete bridges cast in situ, precast reinforced concrete and prestressed bridges, all of which are in use in several Brazilian states and countries internationally.

The analysis methods include the identification of models of small-span bridge design used on rural and neighboring roads; furthermore, a standard model that fulfils the same function to compare environmental and economic performance is proposed.

Regarding the elaboration of bridge typology, research was conducted in several precast industries that produce bridges, and a survey was executed in databases of Brazilian and international transport infrastructure government agencies.

Twenty-seven bridge models were pre-selected, with spans ranging from 6 to 20 m, comprising a total of 405 bridges (Figure 1).

The nomenclature of the identification of the types of transversal bridges was adopted as follows: P01 to P27 represented the 27 types of bridges considered, 2V (two beams) to 6V (six beams) and 8V (eight beams) corresponded to quantities of beams, while M(S) corresponded to mixed welded beams, M(L) to prefabricated laminated beams, VL to in situ, PT to prestressed and PM corresponded to precast, with A, B, C, D and IL corresponding to the companies surveyed.

The following specifications were observed:

A load capacity of 450 kN: the selected roadway was bi-directional, with two traffic lanes with a width equal to 3 m each, with the typical vehicle traffic of the Brazilian standard NBR-7188, the TB-45 (450 kN); an Average Daily Volume (ADV) of 50 < ADV < 200 vehicles per day was considered for the calculation of the number of cycles that occurred in the bridge structure during its lifespan.

Beams: The selected beams were steel-rolled and electro-welded “I” profiles, in addition to pre-cast reinforced concrete and prestressed concrete beams and beams cast in situ.

Slabs: The adopted slabs were made of concrete, precast concrete and cast in situ, with a thickness of 20 cm and a reinforcement ratio that varied according to the quantities of beams in each bridge. Figure 2 presents a schematic model of the bridges analyzed; all bridges consisted of the same types of railings, rainwater drainage and asphalt paving.

To determine the LCCA, the amount of material and equipment hours and the number of personnel hours spent on the execution of each unit were compared and multiplied by the cost of materials, the hourly rent of equipment and the hourly wage of employees, respectively, which duly increased with social charges and budget difference income (BDI). Regarding the equipment, the hourly cost of transportation and operation involving the movement of materials and people within the site included trucks, cranes and breakers, among others. The cost of labor was represented by the wages of the workers who handled the materials, in addition to social charges and other expenses involving workers’ participation.

The raw materials/products considered refer to the services necessary for the execution of the construction extraction stage. The services adopted are part of a list of procedures that begins with the support device, moving through the stages of precast element construction, concreting, asphalt paving, transportation, drains and painting. During the bridge use phase, the transportation services for the visual inspection, cleaning and demolition of asphalt paving as well as transportation to the sorting center (landfill) and pavement replacement, including machinery and equipment, materials and all transportation, were considered.

Finally, in the phase corresponding to the procedures at the end of the bridge life cycle, asphalt paving demolition services were considered. Reinforced concrete and steel beams, when applicable, were transported to the respective sorting centers (landfill). Budgets were prepared including all direct and indirect costs of the bridge construction phases, for all the bridges analyzed; subsequently, cost indicators were created for the small-span bridge superstructures.

A relationship was found between the services considered in reference to the bridge models of the following types: P1—steel/concrete composite bridges; P2—in situ cast reinforced concrete bridges; P3—prestressed concrete beam bridges; and P4—pre-cast reinforced concrete beams bridges. These are presented in Table 1.

Some scenarios were considered to support the decision-making process for the best performing bridge in terms of sustainability. Considering the scenarios, the environmental and economic scores were 50/50, 60/40, 40/60/70/30, 30/70 and 55/45, respectively.

Using life cycle cost analysis (LCCA), the cost of each bridge was calculated, generating a cost per m^2^ of bridge. In the life cycle assessment (LCA), after modeling the ReCiPe method, indices per m^2^ of the bridges in the categories of ecosystem damage, human health and resource depletion were obtained.

To calculate the economic and environmental performance of bridge superstructures, it was necessary to normalize the values of each parameter in order to compare values with the same unit. For that, the average of the values obtained was used as the conventional practice of cost and environmental indicators. The values of each bridge were divided by the mean, thus obtaining the indices for each m^2^ of bridge. In determining the final score, scenarios were considered with the weights shown in Figure 3.

The lowest economic coefficients (costs) and environmental damage (ecosystems, human health and resource depletion) represent the lowest impact in each category and thus represent the best performance of each bridge from the sustainability point of view.

## 3. Results and Discussion

Following the evaluation of the bridge inventories, environmental reports were generated to compare the performance between the analyzed models. In this section, comparisons of the 405 bridges analyzed for each of the three categories of damage at the endpoint are presented.

Figure 4 presents the results of the analyzed ecosystem damage category for all bridges in the study.

The indices highlighted in this comparative assessment of damage to ecosystems confirm that the bridges cast in situ have the greatest impact on the loss of species over a given area, for a certain time, expressed in species per year per meter squared (species.yr/m^2^). Notably, the bridges in reinforced concrete cast in situ of the P9 IL 4V model, which present the worst results in terms of performance in the ecosystems category, range from 1.49 × 10^−5^ species.yr/m^2^ (6 m) to 1.79 × 10^−5^ species.yr/m^2^ (20 m).

The analysis also indicates the best performance for steel/concrete composite bridges with two P1 M(S) 2V welded steel beams, with indexes ranging from 3.52 × 10^−6^ species.yr/m^2^ (19 m and 20 m) to 3.67 × 10^−6^ species.yr/m^2^ (17 m).

Figure 5 shows the results of the impacts on the resources category for all bridges in the study.

Considering the indices highlighted in this comparative assessment, the analysis of the graph indicates how the bridges that most impact the excess cost of future resources production over an infinite period, at US dollars (in 2013) per meter squared (USD 2013/m^2^), are those cast in situ, emphasizing the reinforced concrete bridges cast in situ of the P9 IL 4V model. The worst results were in the category of impact son resources, with a variation from 1.55 × 10^2^ USD 2013/m^2^ (6 m) to 1.73 × 10^2^ USD2013/m^2^ (19 m and 20 m).

The bridges with better performance were highlighted as steel/concrete composite bridges with two welded steel beams, model P1 M(S) 2V. The index for these bridges varied from 1.15 × 10^2^ USD2013/m^2^ (9 m, 10 m, 11 m, 12 m, 13 m, 14 m, 15 m, 16 m, 18 m, 19 m, and 20 m) to 1.17 × 10^2^ USD2013/m^2^ (6 m and 17 m).

Figure 6 presents the results of the human health impact category, analyzed for all bridges in the study.

The graph presents the indices in the comparative assessment. They confirm that the bridges that have the greatest impact on the number of years of life lost and the number of years lived with disability—expressed in DALY per meter squared (DALY/m^2^)—are the bridges cast in situ. Especially, the reinforced concrete bridges cast in situ of the P9 IL 4V model have the worst performance-related results in the category of human health impact, ranging from 3.56 × 10^−3^ DALY/m^2^ (6 m) to 4.19 × 10^−3^ DALY/m^2^ (20 m).

The bridges with the best performance indicated in the study are the steel/concrete composite bridges with two welded steel beams model P1 M(S) 2V, whose indexes varied from 1.63 × 10^−3^ DALY/m^2^ (18 m, 19 m, and 20 m) to 1.73 × 10^−3^ DALY/m^2^ (17 m).

After assessing the damage to ecosystems, human health and resources, we found that the concrete bridges cast “in situ” showed worse environmental performance. This occurs due to high cement consumption, because cement production—especially clinker—is the main factor among all analyzed factors that contribute to the cause of this damage. Composite bridges perform better, mainly due to low cement consumption. Another factor that contributes to this improvement is the type of process that the industry currently uses in steel production (the production of steel comprises about 28% of scrap steel).

The comparative analysis was performed with the application of other methods for the characterization of environmental impacts to validate the results obtained in the ReCiPe methodology. The data obtained with the application of the two methods that are also used in the characterization of environmental impacts are presented: the European methods IMPACT 2002 + V2.14 and ILCD 2011 Midpoint + V1.10, and the American method BEES + V4.07 USA.

Figure 7 presents the final single score results for the IMPACT 2002 method.

Assessing the results generated by the IMPACT 2002 + 12.14 method, the lowest single score indicator for the entire life cycle of the bridge superstructure was shown to be the steel/concrete composite bridge model P1 M(S) 2V with two steel beams. The results presented by this method indicate the same bridge modeled in the ReCiPe methodology causes less environmental impact.

Figure 8 presents the final single score results for the European ILCD 2011 Midpoint + V1.10 method.

Analyzing the data presented in the graph generated by the ILCD 2011 + V1.10 method, the two-beam steel/concrete composite bridge model P1 M(S) 2V has the lowest single score indicator. The results displayed by this method indicate the same P1 M(S) bridge modeled in the ReCiPe methodology causes the least environmental impact.

Figure 9 presents the final single score results for the BEES + V4.07 USA method.

Interpreting the graph data generated by the BEES + V4.07 USA method, the steel/concrete composite bridge model with two P1 M(S) 2V welded steel beams exhibits the lowest single score indicator. The results shown by this method indicate that the P1 M(S) 2V bridge modeled by the ReCiPe methodology also causes less environmental impact.

After the comparison of the results obtained by the methods is applied to the life cycle inventory of the bridge superstructures, it is possible to confirm the results indicating that the steel/concrete composite bridge with two P1 M(S) 2V steel beams causes less impact on the environment. In Figure 10, the cost indices of each bridge model per m^2^ in the entire life cycle are presented; i.e., the sum of the costs of Phases 1, 2, and 3.

According to the indices presented in Figure 10 regarding the total cost, the bridge of the steel/concrete composite model with P4 M(L) 2V rolled beams is apparently the cheapest for the 6 m (€ 540.63/m^2^) and 7 m spans (€ 552.87/m^2^); however, the precast P23 PMD 5V model becomes the most accessible from the 8 m to 20 m span, with a variation in cost from € 534.49/m^2^ (8 m) to € 490.60/m^2^ (17 m). Regarding the highest cost, the P19 PMB 6V precast model is superior for the spans ranging from 6 m to 8 m, with its value varying from € 695.79/m^2^ (6 m) to € 666.61/m^2^ (8 m). For the 9 m (€ 651.76/m^2^) and 10 m (€ 642.08/m^2^) spans, the concrete bridge cast in situ P9 IL 4V model is the most expensive. For the other spans ranging from 11 m to 20 m, the P2 M(S) 3V steel/concrete composite bridge model with welded steel beams becomes the most expensive, varying from € 653.49/m^2^ (11 m) to € 866.50/m^2^ (20 m).

The comparisons are presented with the objective of subsidizing the choice of the most adequate typology according to the evaluated criteria, with a recommendation for sustainability indicators in the construction of bridges so that these models can meet the needs of small-span bridges related to rural and neighboring roads.

As shown, the cost of the bridges is compared to the damage categories (endpoint) in the 1 m^2^ functional unit, allowing the visualization of the economic (LCCA) and environmental aspects (LCA). The span analyzed is 10 m, as this is the most used distance for small-span bridges. The comparison of the models based on cost parameters with the category of damage to ecosystems for the 10 m span led to the results presented in Figure 11.

As shown in Figure 11, the model with the lowest bridge cost for the 10 m span is the P23 PMD 5V five-beam model, with values in the order of € 502.12/m^2^; its ecosystem damage indicator is 4.20 × 10^−6^ species.yr/m^2^. Assessing the lowest indicator of ecosystem damage, the P1 M(S) 2V steel/concrete composite bridge model with two welded steel beams presents damage results in the order of 3.57 × 10^−6^ species.yr/m^2^ with a cost of € 554.96/m^2^.

The cost and environmental parameter differences between the two models are 9.52% and 15.07%, respectively. The P9 IL 4V cast in situ bridge model presents the worst scenario, due to both the higher cost and the fact that it presents the highest indicator of damage to ecosystems in the order of € 642.08/m^2^ and 1.58 × 10^−5^ species.yr/m^2^ for cost and damage, respectively.

Comparing the models in the cost parameters by the resource depletion category for the 10 m range, the results are presented in Figure 12.

As shown in Figure 12, the model that presents the lowest cost for the bridges with a 10 m span is apparently the precast bridge with five beams, P23 PMD 5V, in the order of € 502.12/m^2^, and its resource depletion indicator is 1.26 × 10^2^ USD2013/m^2^. Regarding the lowest resource depletion indicator, the steel/concrete composite bridge model with two P1 M(S) 2V welded steel beams presents depletion results in the order of 1.15 × 10^2^ USD2013/m^2^, at a cost of € 554.96/m^2^. Comparing the two models, the cost difference is 9.52% and the difference in the environmental parameter is 8.92%. The P9 IL 4V cast in situ bridge model presents the worst situation, with the highest cost and resource depletion indicator, in the order of € 642.08/m^2^ for cost and 1.60 × 10^2^ USD2013/m^2^ for depletion.

The results of the comparison of the models regarding the cost parameters combined with the category of damage to human health for the range of 10 m are presented in Figure 13.

The results shown in Figure 13 indicate the P23 PMD 5V precast concrete bridge as the model with the lowest cost between 10 m spans, in the order of € 502.12/m^2^. In addition, the model has as an indicator of human health damage of 1.91 × 10^−3^ DALY/m^2^. Assessing the lowest indicator of human health damage, the steel/concrete composite bridge model with 2 P1 M(S) 2V welded beams gives results in the order of 1.65 × 10^−3^ DALY/m^2^ and a cost of € 554.96/m^2^. Comparing the two models, the cost difference is 9.52% and the difference in the environmental parameter is 13.55%. The P9 IL 4V cast in situ concrete bridge model presents the worst situation, meaning that the cost is higher, presenting a cost of € 642.08/m^2^ and a value of 3.74 × 10^−3^ DALY/m^2^ for depletion.

The results show that the bridge with a span of 10 m that presented the lowest cost was the precast concrete bridge with five beams: smodel P23 PMD 5V. For the same span, the model that presented the best environmental performance in all categories of damage was the steel/concrete composite bridge with two beams: the type “I” welded model P1 M(S) 2V.

The influence of transport distances from the market (city) to work (bridge) was also verified with sensitivity analysis at different distances, calculated for 10 km, 30 km (standard adopted at work), 50 km, 80 km and 100 km. For the LCA, the results indicated an increase of 0.2% per km in terms of the damage to human health, ecosystems and resources; in relation to the LCCA, the costs increased by 0.3% per kilometer.

In the present study, several scenarios of downsizing were analyzed, with economic/environmental scores including 50/50, 60/40, 40/60, 70/30, 30/70 and 55/45.

The results indicate that the scenario presenting a score higher than 50% regarding the economic parameter—the precast P23 PMD 5V concrete bridge with five beams—exhibited the best performance. The scenario with a score higher than 50% regarding the environmental parameter—i.e., the composite bridge P1 M(S) 2V with two electro welded steel beams—showed the best environmental performance.

In this study, two decision-making scenarios were graphically presented; Scenario 1 was applied as 50% for each of the economic and environmental categories, and Scenario 2 was applied with a weight of 60% for the economic and 40% for the environmental categories.

The justification for presenting the 50/50 (sustainable) scenario was motivated by the theory of sustainability, which is evidenced when the damage indices show equal values for the environmental and cost issues. The 60/40 scenario (realistic) was presented taking into consideration the opinion of decision makers, who give more importance to the economic (costs) than environmental issues. This information was gathered from a questionnaire submitted to decision makers.

Figure 14 presents the classification with the best performance considering sustainability, with weights of 50/50 for bridges of 10 m in length.

Analyzing Figure 14, the model that presents the best sustainability performance according to the 50/50 scenario is apparently the steel/concrete composite bridge model with two P1 M(S) 2V welded steel beams, followed by the precast concrete bridge with five P23 PMD 5V type beams, which can be observed to be the two most efficient models.

Note that the precast concrete model with five beams, P23 PMD 5V, was derived from the civil defense kit program of the government of Santa Catarina, where the elements and dimensions were standardized to meet the purpose of the study. The P9 IL 4V in situ cast bridge model shows the worst situation, presenting the worst performance in terms of sustainability assessment in the life cycle of small-span bridge superstructures.

Figure 15 presents the classification with the best performance considering sustainability, with weights of 60/40 for bridges with a 10 m length.

As can be seen Figure 15, the model with the best sustainability performance according to the 60/40 scenario is the precast concrete bridge, P23 PMD 5V; the second-best performing bridge is the steel/concrete composite bridge with two P1 M(S) 2V welded steel beams. Therefore, according to this methodology, these two models can be suggested to be the most efficient considering the sustainability assessment in the life cycle of small-span bridge superstructures.

After applying the proposed scenarios, one can suggest the steel/concrete composite bridge model with two P1 M(S) 2V beams, as shown in Figure 16, and the P23 PMD 5V type precast concrete bridge model, as shown in Figure 17 and Figure 18.

## 4. Conclusions

The study’s main objective was to apply sustainability assessment techniques to the life cycle of small-span bridge superstructures, followed by the proposal of environmental and economic indicators to be integrated in the decision-making process. To achieve this objective, 405 bridges were analyzed according to different categories of damage and sustainability scenarios.

The analysis conducted configured the object of the study with the purpose of finding answers that would make it possible to contribute to the mitigation of the environmental impact, the reduction of resource consumption and the minimization of costs in the life cycle of bridges. The obtained results indicate that, in terms of scenarios presenting a score higher than 50% regarding the economic parameter, the precast concrete bridge with five beams exhibited the best performance. On the other hand, scenarios with a score higher than 50% regarding the environmental parameter pointed to the composite bridge with two welded steel beams as the bridge with the best global performance.

Using the methodological structure of the LCA, the results of the study can contribute to the development of indicators to be applied in small-span bridge designs, assuming that considering the sustainability of bridge designs could ameliorate impacts on the environment. These impacts are currently seen as the response to damage verified in the ecological, economic and social spheres; the aim of sustainable design is a construction method that analyzes the adoption of existing protocols and the possibilities of a design based on the concept of sustainability.

It can be concluded that the evaluation of the sustainability of structures depends on many variables that can influence decision-making, such as the criteria considered, the weights and the decision-making methods with the various attributes used.

The results indicate that the environmental performance of bridges is highly correlated with the type of bridge material, taking into account the fact that different types of materials or bridges exhibit different levels of environmental performance, indicating the importance of authorities reformulating intervention policies in the decision-making process.

The study of alternatives that can be applied in bridge designs is an important contribution to the civil construction sector and thus also to the satisfaction, by the State, of the requirements presented by the population for the use of small-span bridges; thus, the results found in the research confirm that the use of sustainability indicators in the methodological basis of LCA in bridge design can have a positive impact on sustainable development.

In the authors’ opinion, although a very important step towards the consideration of sustainability in bridge design can be made with the adoption of the indicators presented in this study, additional considerations should be included in future investigations, including, the eventual influence of social impacts on the obtained results according to different scenarios.

## Figures and Tables

**Figure 1 ijerph-17-04488-f001:**
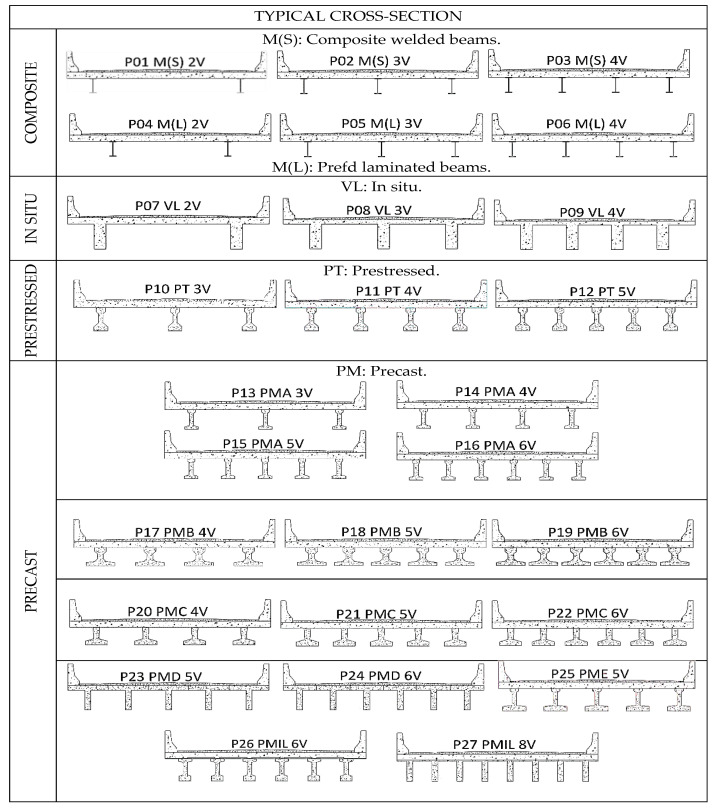
Number of bridges assessed: 27 typologies, 405 total bridges ranging from 6 to 20 meters.

**Figure 2 ijerph-17-04488-f002:**
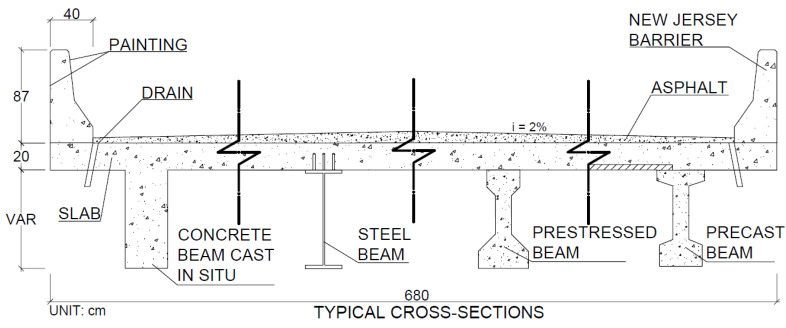
Typical cross-sections of bridge models with concrete beams cast in situ, steel beams, prestressed beams and precast beams.

**Figure 3 ijerph-17-04488-f003:**
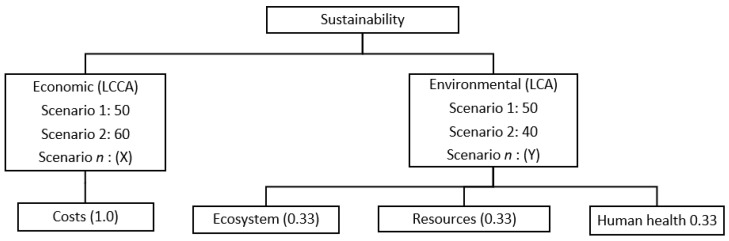
Criteria weights for the assessment of corporate sustainability.

**Figure 4 ijerph-17-04488-f004:**
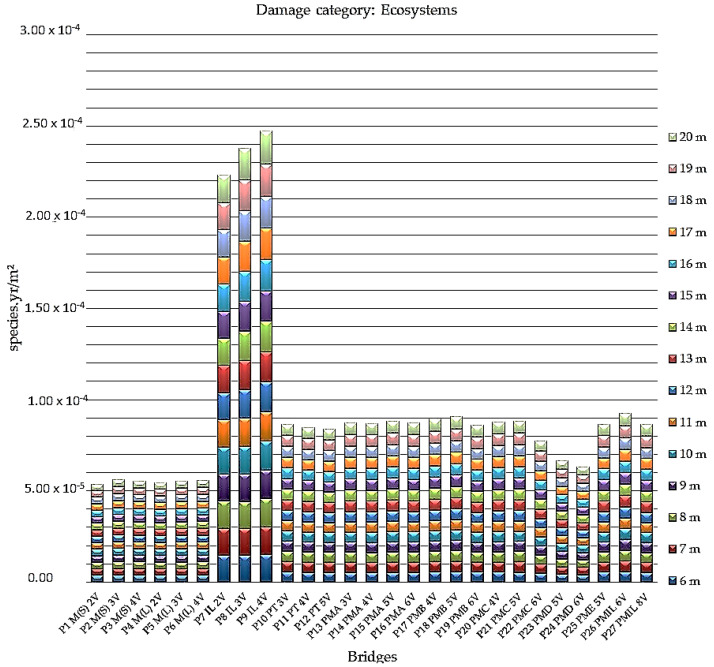
Indices of ecosystem damage in species per year per meter squared for all bridges analyzed.

**Figure 5 ijerph-17-04488-f005:**
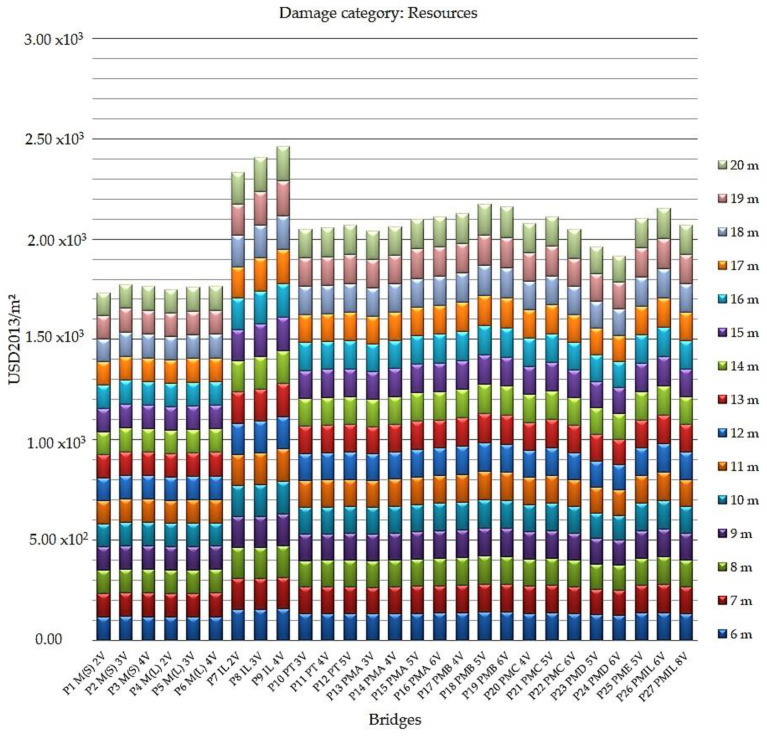
Resources impact indices in US dollars (in 2013) per meter squared for all bridges analyzed.

**Figure 6 ijerph-17-04488-f006:**
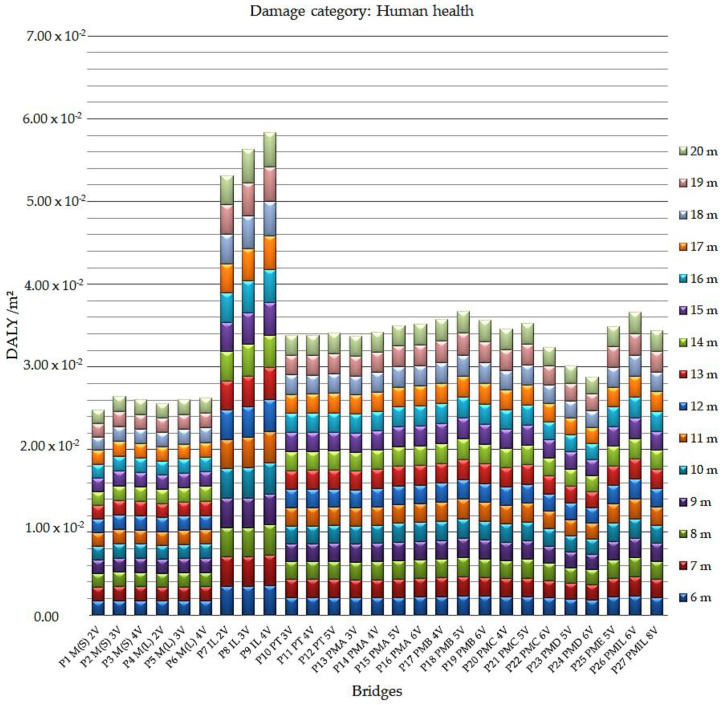
Human health impact indices in Disability Adjusted Life Years DALY per meter squared for all bridges analyzed.

**Figure 7 ijerph-17-04488-f007:**
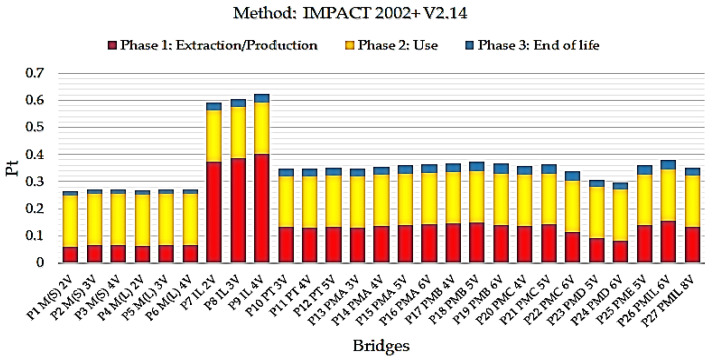
Single scored impact indices for the 2002 IMPACT method for the life cycle of bridges with a 10 m span.

**Figure 8 ijerph-17-04488-f008:**
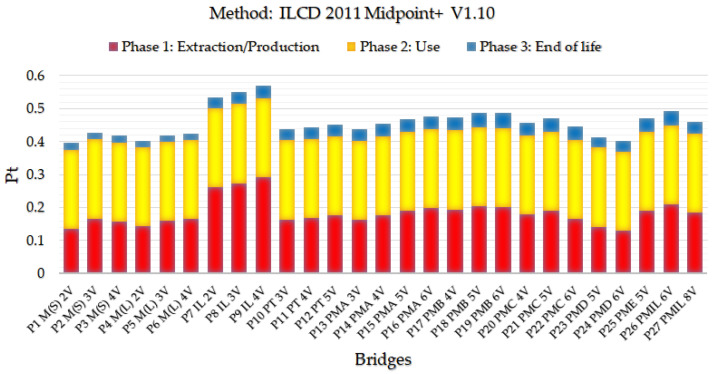
Single score impact indices for the International Reference Life Cycle Data System - ILCD 2011 method for the life cycle of bridges with a 10 m span.

**Figure 9 ijerph-17-04488-f009:**
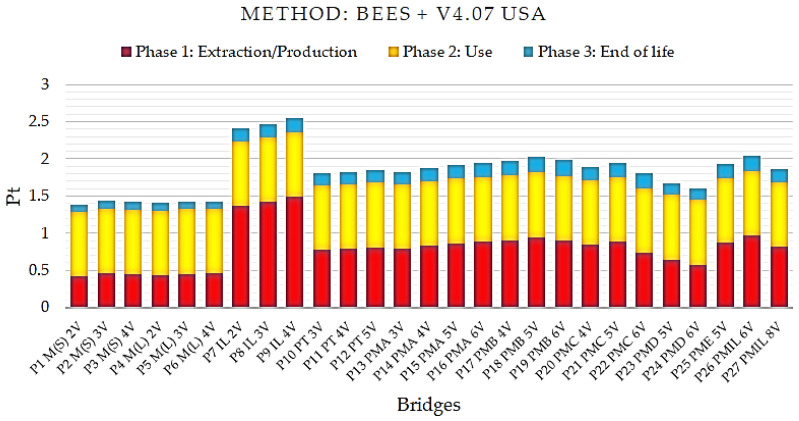
Single scored impact indices for the Building for Environmental and Economic Sustainability - BEES + V4.07 USA method for the life cycle of bridges with a 10 m span.

**Figure 10 ijerph-17-04488-f010:**
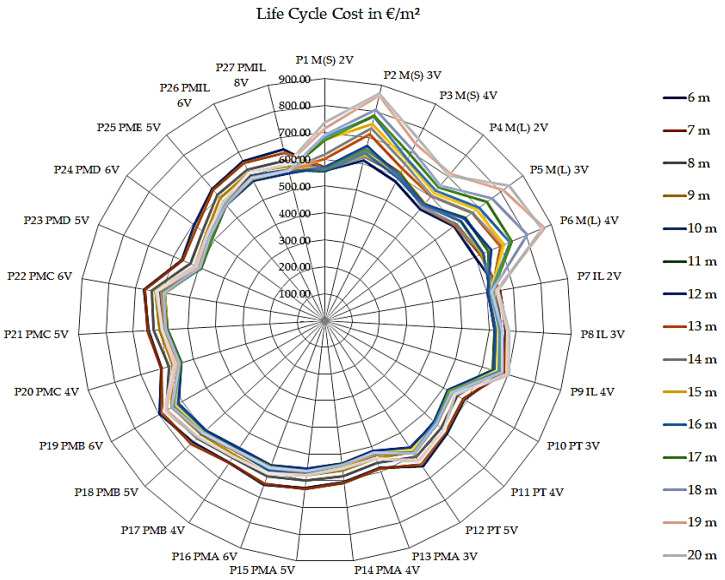
Life cycle cost indices of bridges in EUR per meter squared.

**Figure 11 ijerph-17-04488-f011:**
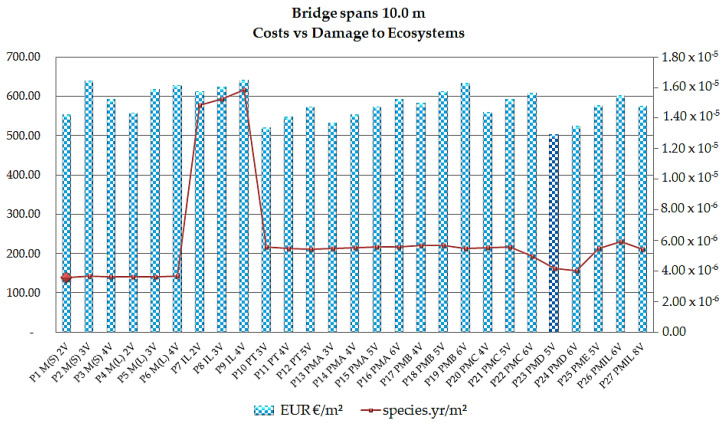
Cost x ecosystem damage indices in EUR/m^2^ and species.yr/m^2^ for the 10 m span.

**Figure 12 ijerph-17-04488-f012:**
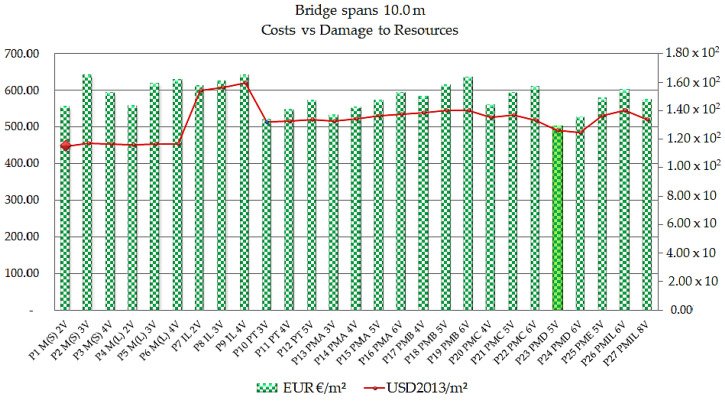
Cost times resources depletion ratios in EUR/m^2^ and USD2013/m^2^ for the 10 m span.

**Figure 13 ijerph-17-04488-f013:**
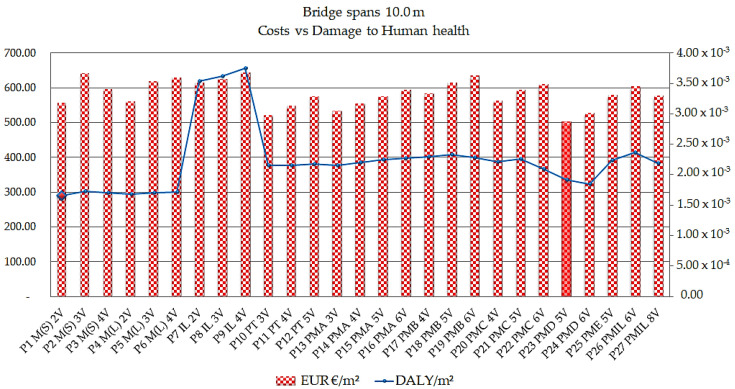
Cost times human health damage indices in EUR/m^2^ and DALY/m^2^ for the 10 m span.

**Figure 14 ijerph-17-04488-f014:**
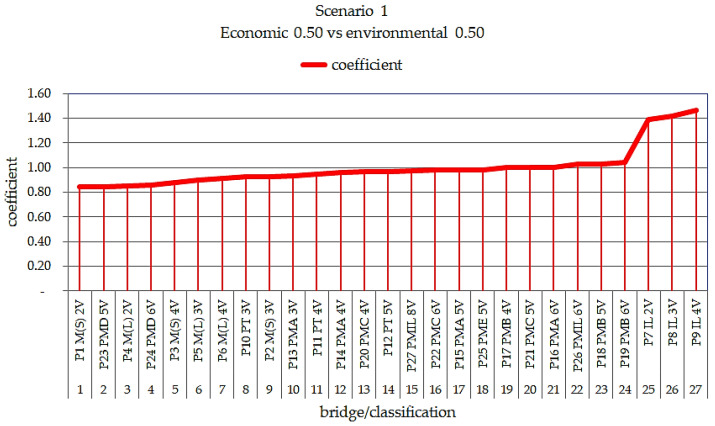
Classification of environmental performance with weights of 0.5 economic and 0.5 environmental per m^2^ for bridges with lengths of 10 m.

**Figure 15 ijerph-17-04488-f015:**
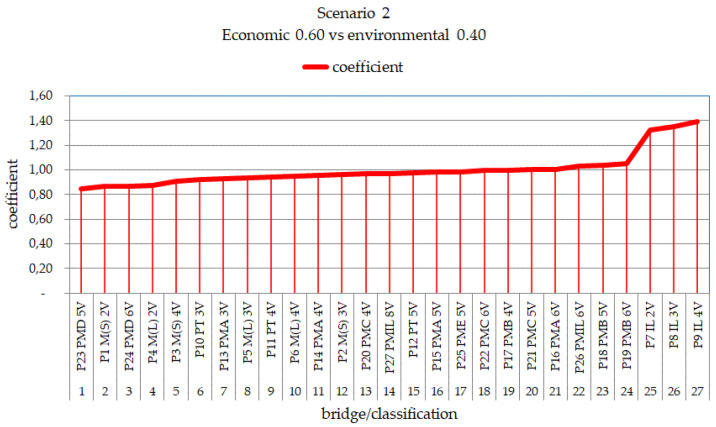
Classification of environmental performance with weights of 0.6 for economic and 0.4 for environmental factors per m^2^ for bridges with lengths of 10 m.

**Figure 16 ijerph-17-04488-f016:**
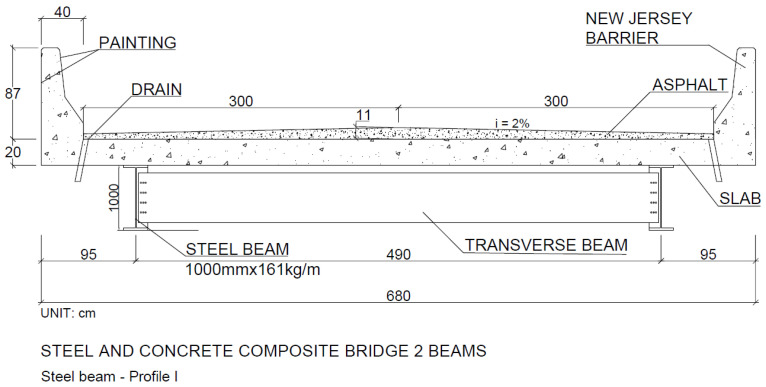
Cross-section of steel/concrete composite bridge P1 M(S) 2V, length 10 m.

**Figure 17 ijerph-17-04488-f017:**
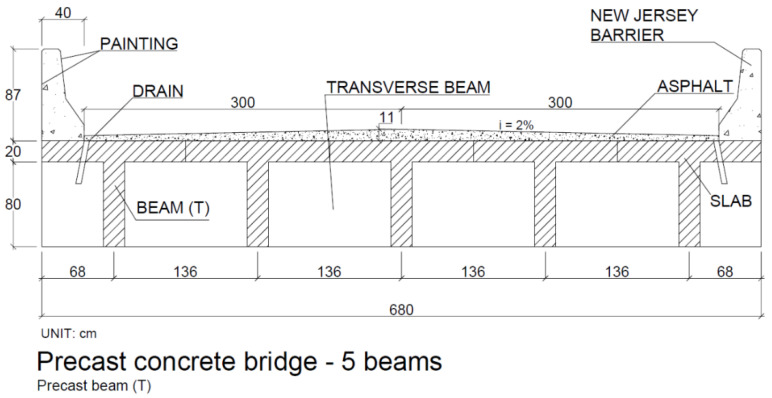
Cross-section of precast concrete bridge, P23 PMD 5V, length 10 m.

**Figure 18 ijerph-17-04488-f018:**
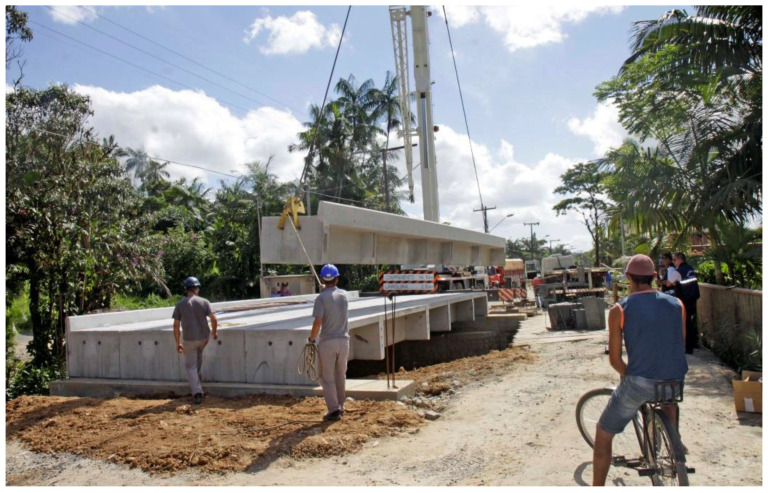
Obstacle transposition kit over Motucas River, at Estrada dos Portugueses, Vila Nova neighborhood, Joinville-SC - Type P23 PMD 5V. Source: Joinville City Hall [75].

**Table 1 ijerph-17-04488-t001:** Products and processes evaluated by bridge type and related to the phases.

Service Description	Unit	P1	P2	P3	P4
Phase (1): Production and construction; Phase (2): use/maintenance; and Phase (3): end of life					
Grout 30 MPa (1)	m^3^	x	x	x	x
Neoprene (1)	dm^3^	x	x	x	x
Concrete production, 35 MPa (sand, basalt, cement, Portland, plasticizer, tap water) (1)	m^3^	x	x	x	x
Concrete mixing factory—construction (1)	p	x	x	x	x
Precast concrete production parts (beams, slab) (1)	m^3^	x	-	x	x
Wood forms (1)	m^3^	x	x	x	x
Steel forms (usage 100 times) (1)	kg	x	x	x	x
Steel rebar (1)	kg	x	x	x	x
Producing I-beams (1)	kg	x	-	-	-
Welding, arc, steel—processing (1)	m	x	-	-	-
Metal working factory—construction (1)	p	x	-	-	-
Hot rolling, steel—processing (1)	kg	x	-	-	-
Building machine (1, 2 and 3)	p	x	x	x	x
Drainage pipes (1)	kg	x	x	x	x
Asphaltic pavement (1 and 2)	t	x	x	x	x
Painting (1 and 2)	m^2^	x	x	x	x
Crane truck (1 and 3)	h	x	-	x	x
Transport (30 km) (1, 2 and 3)	t·km	x	x	x	x
Diesel (1, 2 and 3)	kg	x	x	x	x
Lubricating oil (1, 2 and 3)	kg	x	x	x	x
Electricity, medium voltage (1, 2 and 3)	kWh	x	x	x	x
Industrial machine, heavy, unspecified (1, 2 and 3)	kg	x	x	x	x
Tap water (1 and 2)	kg	x	x	x	x
Inspection (2)	h	x	x	x	x
Pavement demolition (2 and 3)	t	x	x	x	x
Asphalt pavement renovation (2)	t	x	x	x	x
Demolition Building machine (2 and 3)	t	x	x	x	x
Hydraulic digger (1, 2 and 3)	p	x	x	x	x
Waste reinforced concrete (3)	t	x	x	x	x
Treatment of waste asphalt—sanitary landfill (3)	t	x	x	x	x
Treatment of waste reinforcement steel—sorting plant (3)	t	x	-	-	-

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
