# Peer review of "Proposal of Sustainability Indicators for the Design of Small-Span Bridges"

_ijerph, 2020, doi:10.3390/ijerph17124488_

Round 1

Reviewer 1 Report

This paper reports an interesting study pertaining to the appraisal of the sustainability performance of short-span bridge structures, albeit with emphasis on the environmental and economic sustainability dimensions. Also, the use of LCCA (Life Cycle Cost Analysis) and LCEA (Life Cycle Environmental Assessment) techniques respectively. Whereas the implication of the study's finding for decision-making in the contemporary infrastructure delivery system cannot be overemphasized, it will be trite to note that considerable improvements as noted below are necessary to further buttress the significance and credibility of the study's findings. Please note the following observations for your consideration.

  1. The central objective of the study is not clearly articulated in the introductory section of the paper. From the way it was presented, it is difficult to state if the contributions of the paper are in terms of the identification of environmental and economic sustainability indicators affecting decision-making processes during the delivery of sustainable short span bridges OR in the weighting of the various indicators to determine which one contributes most towards negating the sustainability performance of such structures from an environmental impact (destruction of ecosystem), resource consumption, health and cost perspective.
  2. The absence of a credible rationale for neglecting the social dimension of sustainability in the appraisal is indeed worrisome. This is particularly so, considering that the appraisal considered the impact of such bridges on human health-a major facet of social sustainability. 
  3. The section on methods needs to be more explicit. In its current form, the authors work on the assumption that the ReCiPe methodology for transforming life cycle inventory results into a limited number of indicator scores thereby enabling the determination of relative severity on an environmental impact category is one that is known to all the readers. This is not entirely true. Therefore, the methods section will benefit from a comprehensive description of the ReCiPe methodology as deployed in the study. The brief discussion of this methodology in the introductory section will not suffice and the rationale behind its adoption will prove beneficial.
  4. Certain sentences in the text require modification to assist with improved comprehension. For instance, looking at the sentence situated between lines 356 and 359, it is difficult to ascertain if the authors are referring to the results from this study or results from the previously mentioned studies that have deployed various decision-making techniques. 
  5. There is need for a justification of the selection and deployment of the AHP and the VIKOR method in the study as appropriate decision-making tools. Also, a rationale for the use of different weights across two decision-making scenarios is required.

Author Response

The answer to Reviewer #1 is included in an attached pdf file.

Reviewer 2 Report

The main aim of the study was to propose environmental and economic indicators for small span bridges to be integrated in the decision-making process, which is essential part of managing of transportation infrastructure. The reviewer finds the general idea of the paper to be very interesting, as well as very important and current, because of growing problems of infrastructure deterioration.

However, the opinion of the reviewer is that a paper needs a major-revision, and is not acceptable in the current form. More precisely:

1) The main problem of understanding the results presented and conclusions brought is that no database of bridges is presented. It is unknown where the bridges are regarding both macro and micro-locations, and except for the type of superstructure, no other parameters were presented. To that end, no repeatability of analysis by other researchers is possible, as well as no evaluation of results that are presented here. In conclusion, a detailed database of the bridges as well as analysis procedure has to be presented in the beginning of the paper.

2) There are many terms and phrases, which are not clearly defined. Although they may be familiar to the authors, they have to be clearly defined and explained in the paper. The aim of the paper should be to serve also to scientific community, but primarily to be clear to infrastructure managers and operators.

For example:

LINE 45: What exactly are the pillars of sustainability? Please define this in the paper before using it, since this is not a common term in bridge terminology. In addition, what are social pillars?

LINES 134-146: Things like codes, ILCD guidelines etc., as well as different software should all be cited, so that the readers know where to find information for these. The reader is not supposed to be familiar with specific less known software, or to know what does this software do. It should all be (at least shortly) explained in the paper. The same is applied to European and American methods, which should be elaborated in more detail.

LINE 170: Define ADV

FIGURE 2: Why is the AASHTO bridge model used; are the bridges in USA? What does this model show, and why is it presented? Does it want to say that all the analysed bridge types have the same upper structure of the bridge?

LINE 208: Damage categories should be defined and explained, as well as their units.

3) The language and the formulation of sentences should be more precise, especially in introduction.

For example:

LINES 31-35: Unclear sentence. This refers especially to the end of the sentence. E.g. it is not clear to what the word "its" refers. Please revise.

LINES 36-40: Please use commas in sentence, to divide the sentence logically.

LINES 42-43: Please revise the sentence: “Most of this total...”

LINE 154: If the paragraph is connected to the previous one then it should not be separated. If not, then it should not begin with the word “Next,..”.

FIGURE 1: the type of bridges should be called composite rather than mixed, since in the introduction these are called composite.

LINE 169: Please reformulate this sentence since it is not clear to the reviewer. The single carriageway/roadway can have more lanes. However, a single lane cannot have two lanes.

LINES 180-203: The text is divided in 7 small paragraphs. Maybe it should be considered to reduce the number of paragraphs.

The reviewer is very eager to read again the paper in its improved version.

Best regards!

Author Response

The answer to Reviewer #2 is included in an attached pdf file.

Reviewer 3 Report

The manuscript presents the application of techniques for sustainability analysis in the life cycle of small-span bridge superstructures. The aim is to obtain environmental and economic indicators to be integrated into the decision-making process to minimize the environmental impact, reduce resources consumption, and minimize life cycle costs.

The reviewer does not suggest publishing this manuscript in the present format. The overall quality of the manuscript should be improved. Thus, enabling the reviewer an adequate evaluation of the work presented. A major revision is required before the manuscript can be recommended for publication in this journal.

General comments:

The novelty of a research work, with respect to the state-of-the-art in some specific topic, is the major reason for accepting a manuscript for publication in a scientific journal. Although the authors present 60 references, the introduction do not provide the readers with information regarding the novelty of the work reported in the manuscript. The authors should redo the Introduction in order to: adequately present the problem to be solved; present the state-of-the-art identifying an existent the knowledge gap; propose a novel method/technique/strategy to fill the knowledge gap or contribute to improvements in the state-of-the-art. The current Introduction discourage readers to advance in reading the rest of the manuscript.

The information presented in lines 352 to 367 is wrongly placed in Section 3. It Should be moved to the Introduction.

The authors should avoid long sentences. In the manuscript, there are several examples of paragraphs comprising only one sentence.

The title indicates “proposal of sustainability indicators”. Therefore, it is expected that a survey on usual sustainability indicators will be presented. It is also expected that the best/ the most adequate sustainability indicators are used. A discussion on the advantages and disadvantages of using different sustainability indicators should be presented. Furthermore, it can be expected that the authors propose some new sustainability indicators. None of these issues are addressed.

Section 2 should be improved. The authors should provide detailed information regarding the methodologies used, the options considered in the calculations, the values of unit costs and environmental impact of the construction materials considered. The authors should also provide references concerning: ISO 14040, ISO 14044, ILCD guidelines, Ecoinvent database, ReCiPe method, SimaPro software, the European methods, the American method.

The authors should clarify why they select the indices presented in the manuscript instead of other possible indices.

The results obtained for the different indices should be analysed and discussed. For example, the authors should justify why the steel-concrete composite bridges present the best performance for the ecosystem damage and the cast “in situ” reinforced concrete bridges present the worst performance.

The authors should clarify why they selected the 50/50 and the 60/40 scenarios. Why not, 40/60, 70/30, 30/70, 55/45? Probably the results obtained will be different.

Section 4 should be improved. The text presented do not provide relevant conclusions about the work.

Some specific comments:

Title – Perhaps “project” should be replaced by “design”. The authors should think and decide what word is more appropriate.

Please replace “in loco” by “in situ”.

Please replace “resource” by “resources”, where applicable.

Line 23 – Please correct “selection and configuration” by “selection and bridge configuration”

Line 23 – Please correct “configuration, and allowed” by “configuration. Thus, allowing”

Figure 1 – The nomenclature adopted for identifying the cross-sectional bridge types should be described in the text.

Line  170 – ADV is not defined.

Line 174 – Please correct “rate” by “ratio”.

Please replace “metal beams” by “steel beams”.

Line 174 – The authors refer that the slab thickness is 20 cm. The authors should clarify why in Figure 2 a varying thickness of the slab is indicated.

If Figure 15 refer to the best solution for bridges with 10m span the height of the steel longitudinal beam should be clearly defined. The same comment applies to the slab thickness.

Perhaps it is better to replace the image in Figure 16 by a drawing of the bridge cross-section similar to the one presented in Figure 15.

Author Response

The answer to Reviewer #3 is included in an attached pdf file.

Round 2

Reviewer 1 Report

The manuscript has witnessed considerable improvement having incorporated most of the reviewer's comments. 

Author Response

The authors are very grateful for the positive comments of the reviewer. His/her comments have certainly served to improve the quality of the article.

Reviewer 2 Report

The authors offered a comprehensive corrections of the paper and reviewer accepts all the changes.

Author Response

(The authors gave the same response as above.)

Reviewer 3 Report

The authors addressed several reviewer’s comments which improved the overall quality of the manuscript. However, before the manuscript is ready for publication there are several revisions that should be made concerning the following issues:

#1 – In general comment #1 of the previous revision it is stated that: “The novelty of a research work, with respect to the state-of-the-art in some specific topic, is the major reason for accepting a manuscript for publication in a scientific journal. Although the authors present 60 references, the introduction do not provide the readers with information regarding the novelty of the work reported in the manuscript. The authors should redo the Introduction in order to: adequately present the problem to be solved; present the state-of-the-art identifying an existent the knowledge gap; propose a novel method/technique/strategy to fill the knowledge gap or contribute to improvements in the state-of-the-art. The current Introduction discourage readers to advance in reading the rest of the manuscript”. The authors only partly addressed the reviewer comment. They added a minor improvement regarding the novelty of the work reported in the manuscript. As previously stated, the Introduction discourage readers to advance in reading the rest of the manuscript. There is a lack of quality in terms of organization of ideas within the text and in terms of English writing. This is the major drawback of this manuscript and was not significantly improved from the previous version. Section 1, Section 2 and Section 4 should be carefully and entirely reviewed to improve the manuscript quality.

#2 – In general comment #5 it is stated that: “Section 2 should be improved. The authors should provide detailed information regarding the methodologies used, the options considered in the calculations, the values of unit costs and environmental impact of the construction materials considered. The authors should also provide references concerning: ISO 14040, ISO 14044, ILCD guidelines, Ecoinvent database, ReCiPe method, SimaPro software, the European methods, the American method”. The authors addressed this comment providing more information on the methodologies used and several missing references. However, the detailed information presented in lines 181 to 195 should be placed in Section 2.

#3 – The information presented in the first paragraph of the answer to general comment #6 should be referred in the text.

#4 – In general comment #9 it is stated that. “Section 4 should be improved. The text presented do not provide relevant conclusions about the work”. The authors added some sentences in the revised version of the manuscript. However, the overall quality of the conclusion was not significantly improved. In Section 4, the authors present a set of general ideas that can be used almost in any other work. The conclusions should be specific to the topic addressed in the manuscript, presenting the main findings, practical recommendations, and pointing out some future research developments. Moreover, the authors should revise Section 4 paying attention to the English.

#5 – The authors replaced “project” by “design” in the manuscript title. However, the word “project” is used several times in the text and, perhaps, “design” should be used instead, were appropriate. For example, lines 37, 101, 146, 518, 519, 522, 530 and 533.

#6 – The authors replaced “metal beams” by “steel beams” in the revised version of the manuscript. However, the word “metallic” in line 266 and the word “metal” in line 457 should be corrected.

#7 – The authors corrected the slab thickness in Figure 2. Given that, this Figure represents the typical bridge deck cross-section it should be improved providing information regarding the different types of beams considered. In the current version of the Figure, it seems that the beams can be only prestressed or precast.

#8 – Line 14 – Please replace ”was” by “is”.

#9 – Line 18 – Please replace ”reinforced concrete bridges cast in situ” by “cast in situ reinforced concrete bridges”.

#10 – Line 18 – Please replace ”and precast reinforced and prestressed concrete bridges” by “precast bridges and prestressed concrete bridges“.

#11 – Line 19 – Please replace ”were” by “are”.

#12 – Line 22 – Please replace ”indicated” by “indicate”.

#13 – Line 29 – The first sentence of the Introduction should be rephrased. The reviewer suggests: ”Bridges have a major role in the transportation infrastructure, supporting highway traffic loads, crossing various obstacles and performing an effective communication between two destinations”.

#14 – Line 61 – The authors refer ”bridges composed of steel and concrete” and this may refer to almost all types of bridges (steel bridges, reinforced concrete bridges, prestressed concrete bridges, steel-concrete composite bridges, …). The sentence should be improved.

#15 – The sentence in lines 62 to 64 should be rephrased. In the way it is written seems that only in steel-concrete composite system the characteristics of the two materials are combined.

#16 – Line 153 – The sentence ends and the idea is not concluded.

#17 – Line 154 – Perhaps, the word “In” is missing at the beginning of the sentence.

#18 – Line 176 – Please correct ”ISOs 14040 and 14044” by “ISO 14040 [65] and ISO 14044 [66]”.

#19 – Line 196 – The authors refer that they perform a sensitivity analysis using the European methods (IMPACT 2002+V2.14 and ILCD 2011 Midpoint + V1.10) and the US method (BEES + V4.07 USA). After that, in line 349 the authors start presenting the results obtained applying the referred methods. A sensitivity analysis aims to evaluate how a given variable affects the response of a certain numerical model. This was not the case of what is presented in the manuscript. It seems that the authors perform a comparative study and not a sensitivity analysis. This should be corrected.

#20 – Line 246 – The reviewer suggests “Brazilian states and international countries”.

#21 – Line 251 –If the term “national” refer to “Brazilian” this should be corrected.

#22 – Table 1 should be improved. The reader cannot understand what corresponds to Phase 1, 2 or 3.

#23 – Line 296 – Please replace ”bridges in reinforced concrete cast in situ” by “cast in situ reinforced concrete bridges”.

#24 – Line 299 – The caption of Table 1 does not provide an adequate description of what is presented in the Table and should be corrected.

#25 – Line 302 – Please replace ”elaborated” by “considered”.

#26 – Line 346 – Please replace ”are the main factors” by “is the main factor”.

#27 – Line 348 – The authors refer that the better performance of steel-concrete composite bridges is mainly due to the use of recycled steel. In order to improve the results presented, the reviewer suggests that the authors perform the analysis considering non-recycled steel.

#28 – Line 354 – Please correct ”IIMPACT” by “IMPACT”.

#29 – Line 462 – Please replace ”scenarios” by “scenario”.

#30 – Figure 13 and Figure 14 – The authors should refer in the text how the “coefficient” presented in the graphs is calculated. It should also be referred that a lower coefficient is better.

#31 – Lines 502 and 503 – The reviewer suggest removing the sentence because does not add relevant information.

Author Response

The authors addressed several reviewer’s comments which improved the overall quality of the manuscript. However, before the manuscript is ready for publication there are several revisions that should be made concerning the following issues:

#1 – In general comment #1 of the previous revision it is stated that: “The novelty of a research work, with respect to the state-of-the-art in some specific topic, is the major reason for accepting a manuscript for publication in a scientific journal. Although the authors present 60 references, the introduction do not provide the readers with information regarding the novelty of the work reported in the manuscript. The authors should redo the Introduction in order to: adequately present the problem to be solved; present the state-of-the-art identifying an existent the knowledge gap; propose a novel method/technique/strategy to fill the knowledge gap or contribute to improvements in the state-of-the-art. The current Introduction discourage readers to advance in reading the rest of the manuscript”. The authors only partly addressed the reviewer comment. They added a minor improvement regarding the novelty of the work reported in the manuscript. As previously stated, the Introduction discourage readers to advance in reading the rest of the manuscript. There is a lack of quality in terms of organization of ideas within the text and in terms of English writing. This is the major drawback of this manuscript and was not significantly improved from the previous version. Section 1, Section 2 and Section 4 should be carefully and entirely reviewed to improve the manuscript quality.

A.: The authors acknowledge the reviewer for the opportunity to improve the paper. In order to attempt to follow the reviewer’s suggestion, the paper was restructured, with a special attention to Section 1. The ideas were reordered to stress the main objective of the study, and some paragraphs related to general concepts were passed to Section 2.

#2 – In general comment #5 it is stated that: “Section 2 should be improved. The authors should provide detailed information regarding the methodologies used, the options considered in the calculations, the values of unit costs and environmental impact of the construction materials considered. The authors should also provide references concerning: ISO 14040, ISO 14044, ILCD guidelines, Ecoinvent database, ReCiPe method, SimaPro software, the European methods, the American method”. The authors addressed this comment providing more information on the methodologies used and several missing references. However, the detailed information presented in lines 181 to 195 should be placed in Section 2.

A.: The authors agree with the observation and thank the reviewer for his/her recommendation. The paragraphs have been moved to section 2.  Text was transferred to the (207-221)

#3 – The information presented in the first paragraph of the answer to general comment #6 should be referred in the text.

A.: As suggested by the reviewer, we included the paragraph in the article.

Text was transferred to the (LINES 198-201): “In recent years many methods of impact characterization have been developed in the world, as an example we can mention the European methods such as ILCD 2011 MIDPOINT+ and IMPACT 2002+ in NORTH AMERICA: BEES and TRACI 2.1. But the method that has been standing out among researchers is the global ReCiPe method (adopted in this work).”

#4 – In general comment #9 it is stated that. “Section 4 should be improved. The text presented do not provide relevant conclusions about the work”. The authors added some sentences in the revised version of the manuscript. However, the overall quality of the conclusion was not significantly improved. In Section 4, the authors present a set of general ideas that can be used almost in any other work. The conclusions should be specific to the topic addressed in the manuscript, presenting the main findings, practical recommendations, and pointing out some future research developments. Moreover, the authors should revise Section 4 paying attention to the English.

A.: The authors agree with the observation and thank the reviewer for his/her recommendation. Some main findings of the study were included in Section 4.

Text now reads (LINE 534): “The analysis conducted configured the object of the study with the purpose of finding answers that make it possible to contribute to the mitigation of the environmental impact, the reduction of resources consumption, and the minimization of costs in its life cycle. The obtained results indicate that to scenarios presenting a score higher than 50% regarding the economic parameter, the precast concrete bridge with five beams exhibited the best performance. On the other hand, scenarios with a score higher than 50% regarding the environmental parameter pointed to the composite bridge with two welded steel beams as the one with the best global performance”.

Line 559: “In authors opinion, although a very important step towards the consideration of sustainably on the bridges design can be made from the adoption of indicators presented in this study, additional considerations can be included in future investigations. Among them, the eventual influence of social impacts on the obtained results according to different scenarios.”

#5 – The authors replaced “project” by “design” in the manuscript title. However, the word “project” is used several times in the text and, perhaps, “design” should be used instead, were appropriate. For example, lines 37, 101, 146, 518, 519, 522, 530 and 533.

A.: The authors agree with the observation and thank the reviewer for the recommendation. Throughout the text the word "project" has been replaced by " design".

#6 – The authors replaced “metal beams” by “steel beams” in the revised version of the manuscript. However, the word “metallic” in line 266 and the word “metal” in line 457 should be corrected.

A.: The authors agree and appreciate the suggestion.

Text now reads (LINES 275 and 483): "steel”

#7 – The authors corrected the slab thickness in Figure 2. Given that, this Figure represents the typical bridge deck cross-section it should be improved providing information regarding the different types of beams considered. In the current version of the Figure, it seems that the beams can be only prestressed or precast.

A.: The authors are grateful for the suggestion and improved figure 2 as suggested by the reviewer. We inform that the other beams used in the work are shown in Figure 1.

Figure 2 – Typical cross-sections of bridge models with Concrete beams cast in situ, Steel beams, Prestressed beams and Precast beams.”

#8 – Line 14 – Please replace ”was” by “is”.

A.: The authors thank the observation.

#9 – Line 18 – Please replace ”reinforced concrete bridges cast in situ” by “cast in situ reinforced concrete bridges”.

A.: The sentence was rewritten in order to be clearer.Text now reads (LINES 17-18 and 306): "cast in situ reinforced concrete bridges”

#10 – Line 18 – Please replace ”and precast reinforced and prestressed concrete bridges” by “precast bridges and prestressed concrete bridges“.

A.: The authors thank the reviewer for his suggestion and inform that the sentence was rewritten.

#11 – Line 19 – Please replace ”were” by “are”.

A.: The authors thank the reviewer. Text was corrected.

#12 – Line 22 – Please replace ”indicated” by “indicate”.

A.: The authors thank the reviewer for his suggestion.

#13 – Line 29 – The first sentence of the Introduction should be rephrased. The reviewer suggests: ”Bridges have a major role in the transportation infrastructure, supporting highway traffic loads, crossing various obstacles and performing an effective communication between two destinations”.

A.: The authors agree with the observation and thank the reviewer for the recommendation.

Text now reads (LINES 28-29): “Bridges have a major role in the transportation infrastructure, supporting highway traffic loads, crossing various obstacles and performing an effective communication between two destinations”

#14 – Line 61 – The authors refer ”bridges composed of steel and concrete” and this may refer to almost all types of bridges (steel bridges, reinforced concrete bridges, prestressed concrete bridges, steel-concrete composite bridges, …). The sentence should be improved.

A.: The authors thank the reviewer for the recommendation.

Text now reads (LINES 49-50): “of composite steel and concrete structures (concrete deck and steel profile beams)”

#15 – The sentence in lines 62 to 64 should be rephrased. In the way it is written seems that only in steel-concrete composite system the characteristics of the two materials are combined.

A.: The authors appreciate the observation. The sentence was changed.

Text now reads (LINE 52): “with emphasis on the steel/concrete composite bridges (concrete deck and steel profile beams)”

#16 – Line 153 – The sentence ends and the idea is not concluded.

A.: The authors thank the reviewer’s suggestion and inform that the phrase has been rewritten in order to be clearer.

Text now reads (LINES 169-172): “There is a perception that there is a lack of a standardized bridge life cycle assessment manual that can guide designers in their decision making in choosing the best design, with a goal of minimizing environmental and economic damage, with a focus on sustainability.”

#17 – Line 154 – Perhaps, the word “In” is missing at the beginning of the sentence.

A.: The authors thank the reviewer for his/her observation.

#18 – Line 176 – Please correct ”ISOs 14040 and 14044” by “ISO 14040 [65] and ISO 14044 [66]”.

A.: The authors thank the reviewer for his suggestion.

Text now reads (LINE 86): “ISO 14040 [65] and ISO 14044 [66]”

#19 – Line 196 – The authors refer that they perform a sensitivity analysis using the European methods (IMPACT 2002+V2.14 and ILCD 2011 Midpoint + V1.10) and the US method (BEES + V4.07 USA). After that, in line 349 the authors start presenting the results obtained applying the referred methods. A sensitivity analysis aims to evaluate how a given variable affects the response of a certain numerical model. This was not the case of what is presented in the manuscript. It seems that the authors perform a comparative study and not a sensitivity analysis. This should be corrected.

A.: The authors acknowledge that a comparison was made to validate the data.

Text now reads (LINE 91 and 375): “Comparative”

#20 – Line 246 – The reviewer suggests “Brazilian states and international countries”.

A.: The authors thank the reviewer for his suggestion.

Text now reads (LINE 255): “Brazilian states and international countries”

#21 – Line 251 –If the term “national” refer to “Brazilian” this should be corrected.

A.: The authors thank the reviewer for his suggestion.

Text now reads (LINE 260): “Brazilian”

#22 – Table 1 should be improved. The reader cannot understand what corresponds to Phase 1, 2 or 3.

A.: The authors included numbering in Table 1 to reference the phases with the processes and products.

Text now reads (LINES 309-310): “Phase (1): Production and construction; Phase (2): Use/maintenance and Phase (3): End of life” and “(1, 2 and 3)”

#23 – Line 296 – Please replace ”bridges in reinforced concrete cast in situ” by “cast in situ reinforced concrete bridges”.

A.: The authors thank the reviewer for his/her suggestion.

Text now reads (LINES 17,18 and 306): “cast in situ reinforced concrete bridges”

#24 – Line 299 – The caption of Table 1 does not provide an adequate description of what is presented in the Table and should be corrected.

A.:  The title of Table 1 was changed as suggested by the reviewer.

Text now reads (LINE 310): “Table 1 – Products and processes evaluated by bridge type and related to the phases.”

#25 – Line 302 – Please replace ”elaborated” by “considered”.

A.: The authors thank the reviewer for his/her suggestion.

Text now reads (LINE 312): “considered”

#26 – Line 346 – Please replace ”are the main factors” by “is the main factor”.

A.: The authors thank the reviewer for his suggestion.

Text now reads (LINE 370): “is the main factor”

#27 – Line 348 – The authors refer that the better performance of steel-concrete composite bridges is mainly due to the use of recycled steel. In order to improve the results presented, the reviewer suggests that the authors perform the analysis considering non-recycled steel.

A.: The authors removed the sentence where they stated that “The mixed bridges present better performance, mainly due to the use of steel recycling products.”

            The authors find it interesting to draw up a comparison to determine the influence of recycling on steel production. However, they report that the ecoinvent database was used, which in turn is collected on site by experts from the steel industry. This dataset includes raw material extraction (e.g. coal, iron, ore, etc.) and processing, e.g. scrap, coke making, sinter, blast furnace, basic oxygen furnace, electric arc furnace, rolling mill. The steelmaking processes are shown in the flow diagram. Inputs included in the Life Cycle Inventory relate to all raw material inputs, including steel scrap, energy, water, and transport. Outputs include steel and other co-products, emissions to air, water and land.

            Therefore, the authors find it difficult to develop this comparison in the present work. In order to improve the text, the authors rewrote the sentence.

Text now reads (LINES 371-374): “Composite bridges perform better, mainly due to low cement consumption. Another factor that contributes to this improvement is the type of process that the industry currently uses in steel production (production of steel composed by about 28% scrap steel) [76].”

#28 – Line 354 – Please correct ”IIMPACT” by “IMPACT”.

A.: The authors thank the reviewer for his/her suggestion.

Text now reads (LINES 380 and 382): “IMPACT”

#29 – Line 462 – Please replace ”scenarios” by “scenario”.

A.: The authors thank the reviewer.

Text now reads (LINE 488): “scenario”

#30 – Figure 13 and Figure 14 – The authors should refer in the text how the “coefficient” presented in the graphs is calculated. It should also be referred that a lower coefficient is better.

A.: As requested by the reviewer, the authors included figure 3 and clarified the subject in item 2, Materials and Methods.

Text now reads (LINES 315-328): “At ACC (Economic) the cost of each bridge is calculated, generating a cost per m² of bridge. In the LCA (Environmental), after modelling the ReCiPe method, indices per m² of the bridges in the categories of ecosystem damage, human health and resource depletion are obtained

To calculate the economic and environmental performance of bridge superstructures, it was necessary to normalize the values of each parameter in order to compare values with the same unit. For that, the average of the values obtained was used as the conventional practice of cost and environmental indicators. The values of each bridge were divided by the mean, thus obtaining the indices for each m² of bridge. In determining the final score, scenarios were considered with the weights shown in Figure 3.

Figure 3. - Criteria weights for the assessment of corporate sustainability

            The lowest economic coefficients (costs) and environmental damage (ecosystems, human health and resource depletion), represent the lowest impact in each category, and thus represent the best performance of each bridge from the sustainability point of view.”

#31 – Lines 502 and 503 – The reviewer suggests removing the sentence because does not add relevant information.

A.: The authors thank the reviewer for his suggestion and inform that they excluded the sentence. 

The authors are very grateful for the positive comments of the reviewer. His/her comments have certainly served to improve the quality of the article.
